# Deflection Estimation Model for Prestressed Concrete Slabs with Plastic Inserts Forming Voids

**DOI:** 10.3390/ma15093013

**Published:** 2022-04-21

**Authors:** Mindaugas Zavalis, Mykolas Daugevičius, Aidas Jokūbaitis, Robertas Zavalis, Juozas Valivonis

**Affiliations:** Department of Reinforced Concrete Structures and Geotechnics, Faculty of Civil Engineering, Vilnius Gediminas Technical University, Sauletekio Av. 11, LT-10223 Vilnius, Lithuania; mykolas.daugevicius@vilniuistech.lt (M.D.); aidas.jokubaitis@vilniustech.lt (A.J.); robertas.zavalis@vilniustech.lt (R.Z.); juozas.valivonis@vilniustech.lt (J.V.)

**Keywords:** bubble slab, prestressed concrete slab, deflection, inertia moment, finite element method, numerical model, voided slab

## Abstract

Developed and patented more than 30 years ago, the system of slabs with plastic inserts has become very popular, and it is used all over the world today due to the significantly reduced cost of building construction. Experimental tests have shown that the behaviour of simple bending voided slab structures with plastic inserts during loading is very similar to that of solid slabs. However, their deflection and crack resistance are both slightly inferior to those of solid slabs. When using pretensioned reinforcement, the deflection and crack resistance of voided slabs exceed the above parameters for solid slabs. However, when using plastic inserts to form inner voids in slabs, their cross-section along the span becomes variable. In determining the stiffness of such slab, a problem arises in estimating the moment-of-inertia when the cross-section is variable. To estimate the influence of the voids formed by the plastic inserts on the deflection of prestressed concrete slabs, bending tests of two life-size reinforced concrete slabs were performed. The bending results obtained during the experiment were compared with the results obtained from the numerical model and analytical calculations.

## 1. Introduction

The first hollow-core floor slabs were created in the 1950s. This was an innovative system that became very popular due to its advantages over older solid slab systems. The first voided slabs reduced the cost of building construction and the time required to complete buildings. However, due to tubular voids, the production of slabs could only take place in special factories for reinforced concrete structures, and only one-way slabs could be produced [1]. In 1990, Breuning developed a new system for voided slabs. The newly proposed system was close to the system for solid slabs, but it used less concrete [2,3]. This system was designed for the installation of monolithic two-way slabs [4,5,6,7,8], in which voids were formed by plastic hollow inserts.

Research has identified the main factors influencing the behaviour of voided slabs [9]. One is the type of insert system. Depending on the chosen system of plastic inserts, the form of the inserts, and the arrangement of the inserts in the slabs, the concrete content in the slab can be reduced by 20% to 39%. Consequently, the cross-section and stiffness of the slabs also depend on the type of insert system. Depending on the shape of the insert, the slab moment-of-inertia is not constant and determines the stiffness of the structure. Cube, sphere, and elliptical plastic inserts are currently the most commonly manufactured and offered. These inserts form voids in slabs by removing some of the concrete content from the cross-sectional areas of the slabs under tensile and compressive stresses [9]. Research [9] has also determined that slabs with inserts forming conical voids are less resistant to cracking than slabs with spherical inserts. Due to the reduced concrete in the tensile zone of the slab, cracks in the slab formed at 19% less bending load than in a solid slab. The first crack formed in slabs with spherical inserts at 14.26% less bending load than in a solid slab. The decrease in the cracking limit was due to higher stresses in the tensile reinforcement, due to the thinning of the lower cross-section. The above study showed that despite previous cracking, the conical inserts did not have a significant effect on the bearing capacity. Investigations show that the cracking of concrete sections involves reduced stiffness and increase deflection of reinforced concrete constructions [10]. Consequently, the load-bearing capacity of slabs with conical inserts was the same as that of the solid slab, and the load-bearing capacity of slabs with spherical inserts was 13.15% higher than that of the solid slab and slabs with conical inserts. Under the same maximum load, the deflection was found to be 21.5% lower in slabs with conical inserts than in solid slabs under the maximum bending load. Under the same load, the deflection in slabs with spherical inserts was 11.54% lower than in the solid slab, and 12.75% higher than in slabs with conical inserts. Thus, conical inserts do not affect the bearing capacity of the slab but increase the stiffness of the slab more than spherical inserts [9]. Another paper [2] described a test, during which it was found that slabs with ellipsoidal plastic inserts withstand a higher load than slabs with spherical inserts. Changing the insert spacing from 70 mm to 25 mm decreases the load-bearing capacity of voided slabs compared to solid ones. The decrease is from 10 to 19% [11].

An evaluation of the results of the field tests already performed showed that the bearing capacity of slabs with plastic void-forming inserts is slightly lower than that of solid slabs [12,13,14,15].

Due to the reduced mechanical properties of voided slabs, various technological variants of slab production have been tested to find the optimal solution to reduce the self-weight of the slab, without losing the load-bearing capacity and stiffness. One way to increase the stiffness and load-bearing capacity of voided slabs is to change the reinforcement intensity. Tests have shown that voided slabs usually crack in the normal section, but hollow core slabs with a reinforcement intensity of 0.52% crack in the diagonal section [16].

In the search for other ways to increase the stiffness of voided slabs while maintaining reduced self-weight, it is advisable to use prestressing reinforcement instead of conventional reinforcement nets. Additionally, losses of prestress should be taken into account to avoid unnecessary reduction of stiffness and damages [17]. According to the results of author’s [18] research, the use of prestressing reinforcement allows increased crack resistance and stiffness for slabs with void-forming inserts, without increasing the cross-section area of the slab. To date, most experimental and numerical simulation tests have been performed with slabs reinforced with conventional reinforcements, while few tests have been performed with prestressing reinforcements. The experiments estimated the influence of the magnitude of the compressive stress caused by prestressing reinforcements on reinforced concrete construction with a partial prestressing ratio (PPR) [5] or degree of prestressing (µ_p_) [19] ranging from 0.0 to 1.0. The study with voided slabs found that the load-bearing capacity of a voided slab with different PPR values ranges from 82% to 85% of the load-bearing capacity of a similar solid slab. The obtained results showed a huge impact of compressive stress on cracking in voided slabs. A decrease in the maximum crack width and the number of open cracks was recorded compared to voided slabs of the same size, reinforced with nets. Raising the PPR value to 0.81 (or less frequently to 0.71) with an increasing number of prestressing reinforcement bars resulted in an increase in the maximum destructive force of the voided slab to 13.9%, a decrease in the deflection under maximum force by 21%, and an increase in crack resistance of about 41.3% [5].

Numerical studies of the slab were performed using a finite element modelling program [20]. The influence of the insert diameter of the slab height ratio (D/H) on the slab stiffness was determined from the numerical modelling results [20]. When the D/H ratio was between 0.67 and 0.80, the weight of the reinforced concrete slab could be reduced by 27 to 34% compared to a solid slab. Voided slabs with a D/H ratio between 0.64 and 0.80 can withstand 87–85% of the breaking load of a solid slab. Maintaining a D/H ratio between 0.64 and 0.80, and increasing the number of prestressing bars (from 0 to 3), increases the bearing capacity from 40% to 73% and reduces deflection from 50% to 70%. In numerical modelling, changing the value of the D/H ratio resulted in a uniform cracking pattern that resembled the letter “X”, wherein the cracks began at the corners of the numerical model slab and intersected at the load application point.

The mentioned research mostly considered the behaviour of voided slabs, but the methodology for calculating the deflection was not provided. The stiffness calculation of solid slabs is quite simple, due to the solid cross-section. However, it is difficult to determine moment-of-inertia in slabs with void-forming inserts because the cross-sectional area is variable. We aimed to develop a methodology for calculating the deflection of voided slabs and to compare calculated deflections with deflections of experimental life-size prestressed voided reinforced slabs and numerical model deflection results.

## 2. Experimental Studies

### 2.1. Test Specimens

For the experimental test, two reinforced concrete floor slabs with pretensioned main longitudinal reinforcement were designed and manufactured. Below is a scheme of the production of the designed reinforced concrete slabs, with the intended arrangement of the elliptical plastic void-forming inserts and reinforcement bars and nets used (Figure 1).

Sample preparation was performed sequentially according to the prepared slab scheme. First, a reinforcing (lower) net (T-1) was placed in the concreting moulds on the plastic blocks to form a 16 mm thick protective layer of concrete. Above the T-1 net, four reinforcement bars of 12-mm-diameter were placed and anchored into each slab (Figure 2). During production of the slabs, the main bars were prestressed at 530 MPa ± 7%. The initial prestressing load was reduced, taking into account the prestressing losses.

After the anchoring of the reinforcements, plastic inserts were placed in the moulds of slabs, arranged in three rows of thirteen inserts in one row. The inserts were inserted into the longitudinal bars of the lower reinforcing net T-1. A second (upper) reinforcing net (T-2) was placed on the plastic inserts. The upper and lower reinforcement nets were interconnected by stirrups to maintain the design position of plastic inserts during concreting. For the T-1 and T-2, reinforcing nets were used with a 4-mm-diameter smooth surface S500 class reinforcement.

The technical data of the two reinforced concrete floor slabs with pretensioned reinforcements intended for the study are given in Table 1, and the prepared slab is shown in Figure 3.

### 2.2. Material

In both reinforced concrete slabs prepared for testing, voids were formed by elliptical plastic inserts, Cobiax SL-M-160-180, manufactured by “Cobiax” in Cobiax International GmbH, Bielefeld, Germany (Figure 4). The inserts consisted of two shells that were assembled before concreting.

Using plastic inserts of the selected size, the concrete content was reduced by 27.89% compared to the concrete content requirement of a solid slab. The preliminary technical characteristics of plastic inserts are given below (Table 2).

A concrete mix of class C30/37 was used for the production of the slabs. During the production of prestressed concrete slabs, concrete cubes (100 × 100 × 100 mm) were formed to determine the mechanical properties of concrete (Figure 5). Concrete cubes were tested under compression static load with servohydraulic test machine D2000 according to [22]. From the obtained compressive strength of the concrete cubes, the cylindrical and tensile strength of the concrete, and the elastic modulus of the concrete, were calculated. Additionally, reinforcement bars were tested under static load with servohydraulic dynamic test machine LFV600 (manufactured by Walter + Bai Testing Machines in Löhningen, Switzerland) according to [23,24]. The determined mechanical parameters of the concrete and reinforcement are presented in Table 3 and Table 4. Additionally, the reduced prestressing loads are listed in Table 4.

### 2.3. Test Methodology

During the experiment, the deflections and strains of the slab were measured in the compressive zone of the slab, in the tensile zone of the slab, and at the level of the working reinforcement of the slab. In the upper part of the slab, six strain gauges measured the compressive strains of the concrete area. Strains in the tensile area of the slab were measured by six strain gauges in the lower part of the slab. In these areas, the strain gauges were mounted in the area of the central insert. Eleven strain gauges were mounted to the slab wall at the level of the main reinforcement, between the load application points (1-1. I-1: I-11 strain gauges at the level of the tensioned reinforcement; I-12: I-17 strain gauges on the top of the slab, in the compressed part of the slab; I-18: I-23 strain gauges at the bottom of the slab, in the tensile part of the slab; I-34: I-41 displacement gauges at the ends of the slabs; I-24: I-33 deflection gauges). At the ends of the slab, displacement gauges were attached to the pretensioned reinforcements to monitor their displacement. The deflections of the slab were recorded with deflection gauges placed on both sides of the slab, spread over the entire length of the slab: four at the supports; four under load transfer points, and two in the centre of the slab (Figure 6).

Prestressed concrete voided slab tests were performed under static loading with the universal testing frame PLF 2MN, according to the four-point bending scheme. The load was transmitted to the slab through two rigid transverse beams at a distance of 1900 mm. The view of the tested slab is shown in Figure 7.

A force-controlled static load test was performed under a constant speed of 0.5 kN/s. Every 5 kN the test was stopped and measurements were recorded.

The slip of reinforcement at the ends of the slab was measured with mechanical strain gauges (I-34: I-41) (Figure 6) with an accuracy of 0.001 mm. Additionally, strains on the surface of the concrete (I-1: I-11) and deflection of the slab (I24: I33) were measured by a linear variable displacement transducer (LVDT) with an accuracy of 0.001 mm (Figure 6).

## 3. Deflection Design Model

An experimental study showed that the development of cracks at certain cross-sectional areas depends on the level of load. Therefore, the calculation of the deflection is divided into stages that correspond to the opening of certain cracks. In the first stage, the deflection is calculated up to the opening of the crack in the cross-section with the void. In the second stage, the deflection is calculated until the crack opens in the solid cross-section (transversal web). In the third stage, the deflection is calculated to the yielding of the tensile reinforcement.

Three cross-sections were distinguished in the slab under consideration (Figure 8). Section I-I was through the weakest part of the cross-section, where the smallest cross-sectional area and moment of inertia are in the place where the perimeter of the void is greatest (Figure 9). Section II-II was in the parts of the slab where the voids did not have a smooth bottom and top surface. In this section, the cross-sectional flanges take on a trapezoidal shape. In section III-III, the cross-section of the slab was solid and rectangular.

The cross-section of the longitudinal fragment along the span is variable. Existing deflection calculation methods estimate the moment of inertia of the cross-section when the cross-section of the beam or slab along the span is not variable. Figure 10 shows a scheme for determining the position of section II-II, in which the transformed moment of inertia of the cross-section must be determined. A triangle is formed at the void, perpendicular at $0.5\cdot l_{b}$, and the distance between section II-II and section I-I is $\frac{2}{3}\cdot 0.5\cdot l_{b}$, or from the edge of the void $\frac{1}{3}\cdot 0.5\cdot l_{b}=l_{b}/6$.

The moment of inertia of the uncracked cross-section of the slab, when the cross-section is variable along the section III-III and I-I, is determined by the equation:(1)$$I_{red.c}=\left(I_{red.II}\cdot \frac{0.5\cdot l_{b}}{0.5\cdot b_{w.t}+0.5\cdot l_{b}}+I_{red.III}\cdot \frac{0.5\cdot b_{w.t}}{0.5\cdot b_{w.t}+0.5\cdot l_{b}}\right)\cdot k_{shape}$$
where $I_{red.II}$ is the moment of inertia of the transformed cross-section determined in section II-II about the axis of the centre of gravity of the transformed cross-section; and $I_{red.III}$ is the moment of inertia of the transformed cross-section determined in section III-III about the axis of the centre of gravity of the transformed cross-section. Coefficient $k_{shape}$ evaluates the ratio of the areas of regions ($A_{crv}$ and $A_{tr}$); there, the mixture law is used to determine $I_{red.c}.$ The void quarter was transformed into the rectangular and roundness was not assessed. The approximate value of $k_{shape}$ for this type of void is 0.6.

The position of the centre of gravity of the cross-section highlighted in sections II-II (Figure 11) is determined by the following equation:(2)$$y_{c.II}=\frac{S_{red.II}}{A_{red.II}}$$
where $S_{red.II}$ is the static moment of the transformed cross-section (in section II-II) about the lower edge of the cross-section, and $A_{red.II}$ is the correspondingly reduced cross-sectional area. The mentioned geometrical parameters are determined according to:(3)$$S_{red.II}=b_{f1.II}\cdot h_{f1.II}\cdot \left(\frac{h_{f1.II}}{2}+h_{w.II}+h_{f2.II}\right)+b_{w.II}\cdot h_{w.II}\cdot \left(\frac{h_{w.II}}{2}+h_{f2.II}\right)\;+b_{f2.II}\cdot h_{f2.II}\cdot \frac{h_{f2.II}}{2}+a_{II.t}\cdot h_{II.t}\cdot \left(h_{f2.II}+h_{w.II}-\frac{1}{3}\cdot h_{II.t}\right)\;+a_{II.b}\cdot h_{II.b}\cdot \left(h_{f2.II}+\frac{1}{3}\cdot h_{II.b}\right)+\left(\alpha_{p1}-1\right)\cdot A_{p1}\cdot a_{p1.II};\;A_{red.II}=b_{f1.II}\cdot h_{f1.II}+b_{w.II}\cdot h_{w.II}+b_{f2.II}\cdot h_{f2.II}+a_{II.t}\cdot h_{II.t}\;+a_{II.b}\cdot h_{II.b}+\left(\alpha_{p1}-1\right)\cdot A_{p1}.$$

The moment of inertia of the transformed cross-section (II-II) is determined by:(4)$$I_{red.II}=\frac{b_{f1.II}\cdot h_{f1.II}^{3}}{12}+b_{f1.II}\cdot h_{f1.II}\cdot {\left(h_{f2.II}+h_{w.II}+\frac{h_{f1.II}}{2}-y_{c.II}\right)}^{2}+\frac{b_{w.II}\cdot h_{w.II}^{3}}{12}\;+b_{w.II}\cdot h_{w.II}\cdot {\left(h_{f2.II}+\frac{h_{w.II}}{2}-y_{c.II}\right)}^{2}+\frac{b_{f2.II}\cdot h_{f2.II}^{3}}{12}+b_{f2.II}\cdot h_{f2.II}\cdot {\left(y_{c.II}-\frac{h_{f2.II}}{2}\right)}^{2}\;+2\cdot \left[\frac{a_{II.t}\cdot h_{II.t}^{3}}{36}+\frac{1}{2}\cdot a_{II.t}\cdot h_{II.t}\cdot {\left(h_{f2.II}+h_{w.II}-\frac{1}{3}\cdot h_{II.t}-y_{c.II}\right)}^{2}\right]\;+2\cdot \left[\frac{a_{II.b}\cdot h_{II.b}^{3}}{36}+\frac{1}{2}\cdot a_{II.b}\cdot h_{II.b}\cdot {\left(h_{f2.II}+\frac{1}{3}\cdot h_{II.b}-y_{c.II}\right)}^{2}\right]\;+\left(\alpha_{p1}-1\right)\cdot A_{p1}\cdot {\left(y_{c.II}-a_{p1.II}\right)}^{2}.$$

The position of the centre of gravity of the cross-section highlighted in sections III-III (Figure 12) is determined by the following equation:(5)$$y_{c.III}=\frac{S_{red.III}}{A_{red.III}}$$
where $S_{red.III}$ is the static moment of the transformed cross-section (in section III-III) about the lower edge of the cross-section, and $A_{red.III}$ is the corresponding area of the transformed cross-section. The mentioned geometrical parameters are determined by the formulas:(6)$$S_{red.III}=b_{III}\cdot h_{III}\cdot \frac{h_{III}}{2}+\left(\alpha_{p1}-1\right)\cdot A_{p1}\cdot a_{p1.III}.$$
(7)$$A_{red.III}=b_{III}\cdot h_{III}+\left(\alpha_{p1}-1\right)\cdot A_{p1}.$$

The moment of inertia of the transformed cross-section (III-III) is determined by:(8)$$I_{red.III}=\frac{b_{III}\cdot h_{III}^{3}}{12}+b_{III}\cdot h_{III}\cdot {\left(y_{c.III}-\frac{h_{III}}{2}\right)}^{2}+\left(\alpha_{p1}-1\right)\cdot A_{p1}\;\cdot {\left(y_{c.III}-a_{p1.III}\right)}^{2}.$$

By setting the geometrical parameters of the cross-section, the deflection of the slab can be calculated until the cracking moment in the cross-section with the void is reached. Slab deflection from the external load until the moment ($M_{q.crc.I})$ of cracking:(9)$$\omega_{s.q.I}=k\cdot \frac{M_{q.crc.I}}{E_{c}\cdot I_{red.c}}\cdot l_{s}^{2}$$
where the coefficient *k* evaluates the loading type. The cracking moment can be calculated by the equation:(10)$$M_{q.crc.I}=f_{ct}\frac{I_{red.I}}{y_{c.I}}+P_{m}\left(e_{I}+r_{I}\right)$$
where $f_{ct}$ is the concrete tensile strength; *P_m_* is the prestress force; $e_{I}$ is the distance from the centre of gravity of cross-section to the prestress force; and $r_{I}=\frac{W_{red.I}}{A_{red.I}}$ is the core radius.

When the external load reaches the moment of cracking, the stiffness of the slab is significantly reduced due to the open cracks. The increment of deflection becomes faster. First, cracks open primarily at the voids (Figure 13), and as the load increases, they also open at the section III-III position.

Deflection of the slab from the external load, when the slab is cracked at the parts with the voids until new cracks open in the solid cross-section, (transversal web, $M_{q.crc.III}$):(11)$$\omega_{s.q.cr.II}=k\cdot \frac{1}{r_{m}}\cdot l_{s}^{2}.$$

The following equation evaluates the mean curve when the parts with the voids are cracked and parts within the solid cross-section (transversal web) are not cracked:(12)$$\frac{1}{r_{m}}=\frac{M_{q.crc.I}}{E_{c}\cdot I_{red.c}}\cdot {\left(\frac{M_{q.crc.I}}{M_{q.crc.III}}\right)}^{2}+\frac{M_{q.crc.III}}{E_{c}\cdot I_{red.crc.c}}-\frac{M_{q.crc.III}}{E_{c}\cdot I_{red.crc.c}}\cdot {\left(\frac{M_{q.crc.I}}{M_{q.crc.III}}\right)}^{2}$$
where $I_{red.crc.c}$ is the moment of inertia of the cracked cross-section of the slab, when the cross-section is variable along the section III-III and I-I, which is determined by the equation:(13)$$I_{red.crc.c}=\left(I_{red.crc.II}\cdot \frac{0.5\cdot l_{b}}{0.5\cdot b_{w.t}+0.5\cdot l_{b}}+I_{red.III}\cdot \frac{0.5\cdot b_{w.t}}{0.5\cdot b_{w.t}+0.5\cdot l_{b}}\right)\cdot k_{shape}$$
where $I_{red.crc.II}$ is the moment of inertia of the transformed cross-section (II-II) with cracks about the axis with respect to the neutral axis (depth $x_{II}$).

The depth $x_{II}$ of the neutral axis (Figure 14) of the cracked cross-section:(14)$$x_{II}=\frac{-B_{II}+\sqrt{B_{II}^{2}+4\cdot A_{II}\cdot C_{II}}}{2\cdot A_{II}}.$$

Used designations:(15)$$A_{II}=b_{f1.II}\cdot 0.5;$$
(16)$$B_{II}=\left(\alpha_{s2}-1\right)\cdot A_{s2}\cdot \frac{1}{k_{s2.II}}+\alpha_{p}\cdot A_{p}\cdot \frac{1}{k_{p.II}};$$
(17)$$C_{II}=\left(\alpha_{s2}-1\right)\cdot A_{s2}\cdot a_{s2}\cdot \frac{1}{k_{s2.II}}+\alpha_{p}\cdot A_{p}\cdot d\cdot \frac{1}{k_{p.II}}.$$

If $x_{II}>h_{f1.II}$, then the moment of inertia:(18)$$I_{red.crc.II}=\frac{b_{f1.II}\cdot h_{f1.II}^{3}}{12}+b_{f1.II}\cdot h_{f1.II}\cdot {\left(x_{II}-\frac{h_{f1.II}}{2}\right)}^{2}+\frac{b_{w.II}\cdot {\left(x_{II}-h_{f1.2}\right)}^{3}}{12}\;+b_{w.II}\cdot \left(x_{II}-h_{f1.II}\right)\cdot {\left(\frac{x_{II}-h_{f1.II}}{2}\right)}^{2}\;+2\cdot \left[\frac{a_{II.t}\cdot {\left(x_{II}-h_{f1.II}\right)}^{3}}{36}+\frac{1}{2}\cdot a_{II.t}\cdot \left(x_{II}-h_{f1.II}\right)\cdot {\left(\frac{2}{3}\cdot \left(x_{II}-h_{f1.II}\right)\right)}^{2}\right]\;+\alpha_{p1}\cdot A_{p1}\cdot {\left(h_{f1.II}+h_{w.II}+h_{f2.II}-x_{II}-a_{p1.II}\right)}^{2}.$$

If $x_{II}\leq h_{f1.II}$ (Figure 15), then the moment of inertia:(19)$$I_{red.crc.II}=\frac{b_{f1.II}\cdot x_{fII}^{3}}{12}+b_{f1.II}\cdot x_{II}\cdot {\left(\frac{x_{II}}{2}\right)}^{2}+\alpha_{p1}\cdot A_{p1}\cdot {\left(h_{f1.II}+h_{w.II}+h_{f2.II}-x_{II}-a_{p1.II}\right)}^{2}.$$

The cracking moment in section III-III can be calculated by the equation:(20)$$M_{q.crc.III}=f_{ct}\frac{I_{red.III}}{y_{c.III}}+P_{m}\left(e_{III}+r_{III}\right).$$

Deflection of the slab when the slab is cracked at the parts with the voids and in the solid cross-section parts (transversal webs):(21)$$\omega_{s.q.cr.III}=k\cdot \frac{1}{r_{m}}\cdot l_{s}^{2}.$$

The following equation evaluates the mean curve when the parts with the voids and solid cross-section (transversal web) are cracked:(22)$$\frac{1}{r_{m}}=\frac{M_{q.crc.I}}{E_{c}\cdot I_{red.c}}\cdot {\left(\frac{M_{q.crc.I}}{M_{q.u}}\right)}^{2}+\frac{M_{q.crc.III}}{E_{c}\cdot I_{red.crc.c}}\cdot {\left(\frac{M_{q.crc.III}}{M_{q.u}}\right)}^{2}-\frac{M_{q.crc.III}}{E_{c}\cdot I_{red.crc.c}}\cdot {\left(\frac{M_{q.crc.I}}{M_{q.u}}\right)}^{2}\;+\frac{M_{q.u}}{E_{c}\cdot I_{red.crc.u}}\cdot {\left(\frac{M_{q.u}}{M_{q.u}}\right)}^{2}-\frac{M_{q.u}}{E_{c}\cdot I_{red.crc.u}}\cdot {\left(\frac{M_{q.crc.III}}{M_{q.u}}\right)}^{2}$$
where $I_{red.crc.u}$ is the moment of inertia when the voids and solid cross-section (transversal web) are cracked, which is determined by the equation:(23)$$I_{red.crc.u}=\left(I_{red.crc.II}\cdot \frac{0.5\cdot l_{b}}{0.5\cdot b_{w.t}+0.5\cdot l_{b}}+I_{red.crc.III}\cdot \frac{0.5\cdot b_{w.t}}{0.5\cdot b_{w.t}+0.5\cdot l_{b}}\right)\;\cdot k_{shape}.$$

$I_{red.crc.III}$ is the moment of inertia of the transformed cross-section (III-III) with cracks about the neutral axis with respect to the neutral axis (depth $x_{III}$).

The depth x_III_ of the neutral axis (Figure 16) of the cracked cross-section:(24)$$x_{III}=\frac{-B_{III}+\sqrt{B_{III}^{2}+4\cdot A_{III}\cdot C_{III}}}{2\cdot A_{III}}\;.$$

Used designations:(25)$$A_{III}=b_{f1.II}\cdot 0.5;$$
(26)$$B_{III}=\left(\alpha_{s2}-1\right)\cdot A_{s2}\cdot \frac{1}{k_{s2.III}}+\alpha_{p}\cdot A_{p}\cdot \frac{1}{k_{p.III}};$$
(27)$$C_{III}=\left(\alpha_{s2}-1\right)\cdot A_{s2}\cdot a_{s2}\cdot \frac{1}{k_{s2.III}}+\alpha_{p}\cdot A_{p}\cdot d\cdot \frac{1}{k_{p.III}}.$$

Moment of inertia for Figure 16 cross-section:(28)$$I_{red.crc.III}=\frac{b_{III}\cdot x_{III}^{3}}{12}+b_{III}\cdot x_{III}\cdot {\left(\frac{x_{III}}{2}\right)}^{2}+\alpha_{p1}\cdot A_{p1}\cdot {\left(h_{III}-x_{III}-a_{p1.III}\right)}^{2}.$$

Equations to predict depth of the neutral axis contain coefficients $k_{s2.II};\;k_{s2.III}$ and $k_{p.II};k_{p.III}$. These coefficients evaluate slower increments of deformation in sections II-II and III-III. Therefore, the depth of the neutral axis in section II-II is greater than in section I-I. The depth of the neutral axis in section III-III is greater than in section II-II. Coefficients $k_{p.II};k_{p.III}$ were predicted, taking into account developed deformations in the experimental slabs (Figure 17 blue line). Values of these coefficients are depicted in Figure 17 by green dots. For section III-III, the base value is 66 mm, because this is the width of the web. For section II-II, the base value is 166 mm; this is the distance between both sections II-II.

Calculations may be made up to a bending moment that does not exceed the load-bearing capacity.

## 4. Numerical Slab Analysis

Three-dimensional numerical models of the analysed slabs were created by the finite element program Diana IE 10.3 [25,26]. The models were created taking into account the predicted cross-sectional parameters. Instead of a plastic insert, a void was created, and the effect of the plastic insert itself on the slab behaviour was not evaluated. The volume of the slab was meshed with eight-node hexa-quad type finite elements, estimating the desired edge length of the finite element to be 20 mm. The total amount of elements in the first slab (I pl) was 300,537 and in the second slab (II pl) was 309,323. To fill the three-dimensional volume of slabs with finite elements, the program could additionally use tetra-triangle type elements. The developed finite element models are shown in Figure 18. A total strain-based crack model was assigned to the slab concrete to evaluate the development of cracks according to the rotating method. In the FEA, concrete tensile behaviour was evaluated by an exponential tensile curve, which requires tensile strength and tensile fracture energy. The compressive behaviour parabolic compression curve, which requires the compressive strength and compressive fracture energy, was also evaluated.

The experimental slabs were supported on steel supports. Beneath the steel plates were steel hinges, which ensured the degrees of freedom of the freely supported, statically calculated beams. Appropriately modelled reinforced concrete slabs were also supported on steel plates. These support plates can be divided into separate parts to form the middle edges, to which the support conditions can be assigned. The right-hand support was evaluated in the model as mobile hinged support, so its lower centre edge was assigned a displacement constraint only to the vertical *z*-axis. The left-hand support was considered to be hinged support, and its lower centre edge was assigned with a displacement constraint to the vertical *z* and longitudinal *x*-axis. Since the entire volume of the slabs was modelled, the offset of the central edges of the support plates was constrained by the displacement in the transverse direction of the slab to the *y*-axis. Steel support plates and load transfer plates are considered to be bodies of elastic material that meet the elastic characteristics of steel. In the numerical model, the external load is transmitted through steel plates. These plates, like the lower support plates, were filled with hexa-quad type finite elements. In the inner volume of the slabs, the embedded steel reinforcement was evaluated, which consisted of pretensioned reinforcement bars, bottom reinforcement mesh, top reinforcement mesh, and transverse mounting bars. Elasto-plastic work was assigned to all reinforcement elements. The plasticity of reinforcement was evaluated using the Von-Mises criteria.

Three load sets were created for each numerical slab model. The first load set provided prestress to the four main reinforcement bars. The second load set evaluated the self-weight of all elements involved in the model. In the third load set, a pressure corresponding to 1 kN was added to the upper steel plate. The steps of nonlinear numerical analysis were evaluated according to the order of load transfer. In the first stage, a nonlinear calculation was performed by transferring the prestress force to the slab, creating the initial state of stresses in the slab. In the second stage, a nonlinear calculation was performed by estimating the influence of the self-weight of the elements. In the third stage, the slab was loaded in steps with an external load provided pressure. When large displacements were reached, the numerical analysis was terminated because the strength limit was exceeded in the concrete or reinforcement (large displacements occurred).

## 5. Comparison of Tests, Calculations, and Numerical Analysis Tests

The deflections calculated theoretically and by the finite element method were compared with the experimental ones in Figure 19 and Figure 20. During the experiment, the increase of external load was stopped for a particular time period and measurements were taken and recorded. At that time, the slabs were subjected to a constant load for a short time, which resulted in the development of short-term creep, which changed the stiffness of the slabs. When increasing the external load, the deflection increment differed from that calculated theoretically and by the finite element method. The theoretical calculation methodology and the finite element method do not evaluate the effect of short-term creep. The calculated deflections were larger than the experimental ones. The difference between the calculated and experimental deflections was variable and depended on the loading level. The deflections determined according to the proposed analytical method in the first stage (up to the opening of the crack in the cross-section with the void) are from 1.09 to 1.38 times higher than the experimental ones. Correspondingly, the deflections determined by the finite element analysis are greater from 1.29 to 1.99 times. The difference is greater because cracking develops earlier in the numerical model. In the second stage (when the load reaches the level at which the webs begin to crack), deflections determined according to the proposed analytical method are from 1.16 to 1.52 times higher than the experimental ones. Correspondingly, the deflections determined by the finite element analysis are from 1.67 to 1.81 times greater. In the numerical model, the development of cracks at the web edges is negligible. Therefore, the deflection development in the numerical model was almost linear after the first cracking. This may have been influenced by the size and type of finite elements and the concrete mechanical model.

Relationships between bending moments and strains of prestressed concrete slabs were obtained from gauge readings during the experimental tests (Figure 21 and Figure 22). The purpose of the diagrams was to determine how the strain varies in the installation area of the inserts from the walls formed between the inserts.

The diagram of the I-7 gauge (Figure 21a) shows a compression deformation instead of tensile one. The difference between the similar gauges (I-1; I-3; I-5; I-9; I-11) occurred because of crack which opened in the fixing spots of gauge I-7.

As can be seen from the diagrams above (Figure 22), tensile strains occurred in the areas of the inserts and walls at the beginning of the test. Observing the change of strains, it was found that in the elastic stage (when the value of the bending moment was 44% of the maximum bending value) the total change of strain in the wall area of the load slab was up to 1.36 times higher than in the installation area of the plastic inserts. When the load was increased from 44% to 54% of the maximum moment value, the change of strain in the area of installation of inserts changed from elastic to plastic, and the total change of strain in the area of inserts was 2.18 times higher than in the area of walls. The first cracks were recorded in the tensile zone of the slab, where inserts were installed. The plastic strain of the wall area occurred between 54% and 66% of the value of the maximum bending moment. The diagram (Figure 22) shows that plastic strains were recorded at the location of the I-1 and I-3 gauges. At the mounting locations of the I-5, I-9, and I-11 gauges, elastic strains were recorded at up to 87% of the maximum bending moment value. From 87%, the failure of the slab began.

The diagrams of the bending moments average strain values (Figure 23) show that the change of strains in the zone of walls and inserts were elastic. Despite the presence of voids in the insert area, lower strains were recorded than in the wall area. However, due to the amount of concrete in place of the inserts, plastic strains of the concrete layer occurred more rapidly, i.e., development of cracking.

At the bottom of the slab (Figure 24), from 54% of the maximum bending moment value, the gauges I-19 and I-22 showed plastic tensile strain. From 87% of the maximum bending moment value, the other gauges (I-18, I-20, I-21, I-23) showed a constant increment of strain.

At the top of the slab (Figure 25), elastic strains of the concrete were recorded at up to 43% of the maximum value of bending moments, and from 43%, elastoplastic strains were recorded. Comparing the readings of the gauges in the tensile zone (Figure 24) with the readings of the gauges in the compressive zone (Figure 25), it can be seen that the strains in the tensile zone were on average twice as large as in the compressive zone under the same load level.

## 6. Conclusions

The amount of concrete was reduced by up to 28% in the prestressed concrete slabs with plastic elliptical inserts, compared to solid slabs, thus reducing the self-weight of the slab by up to 28%.

A method for the deflection calculation of prestressed concrete slabs with plastic inserts was proposed. This method takes into account variations of the cross-section geometry along the span of the slab. Additionally, uncracked and cracked cross-sections are considered.

The experimental results confirmed that the position of the neutral axis in the cracked sections (at the place of the inserts and between the inserts) was different. Therefore, the proposed method for deflection calculation allows estimation of the existing state of stress-strain in different cross-sections of the slab when different positions of the neutral axis in the cracked cross-sections are determined.

The comparison of theoretical and experimental results showed that the proposed analytical method predicts the deflection of prestressed concrete voided slabs quite well up to the load level of 80% of failure load.

## Figures and Tables

**Figure 1 materials-15-03013-f001:**
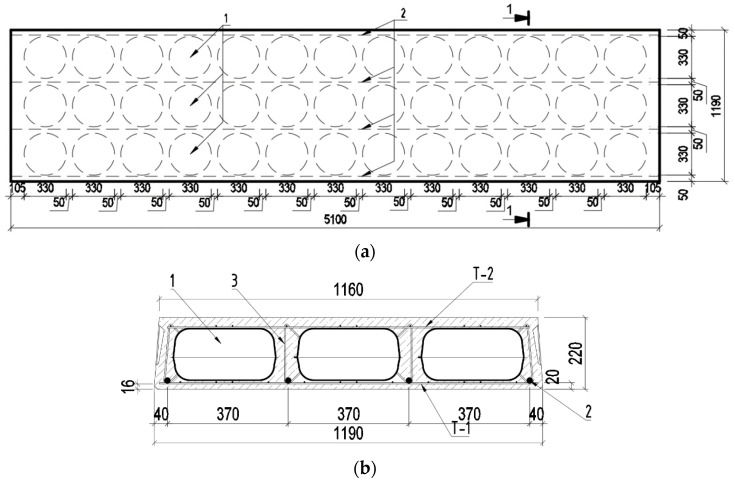
Designed reinforced concrete slab. (**a**) General scheme of the slab, (**b**) slab cross-section 1-1; 1. Plastic insert; 2. Pretensioned reinforcement bars; 3. Stirrup to tie the reinforcement nets; T-1 and T-2 upper and lower reinforcement net.

**Figure 2 materials-15-03013-f002:**
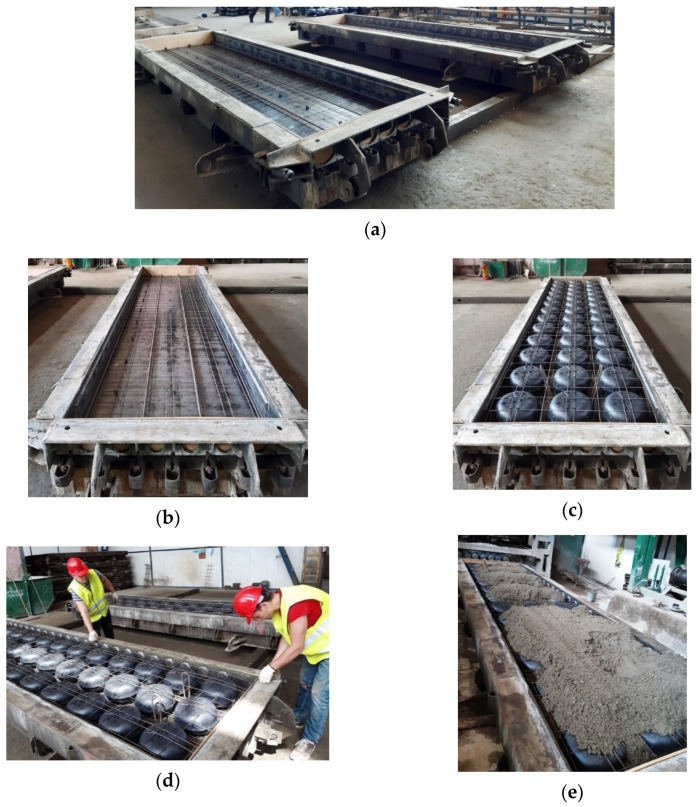
Manufacturing of tested slabs. (**a**) Slab moulds, (**b**) mould with lower slab reinforcement net, (**c**) mould with void-forming inserts and reinforcement, (**d**) moulds prepared for concreting, (**e**) concreting.

**Figure 3 materials-15-03013-f003:**
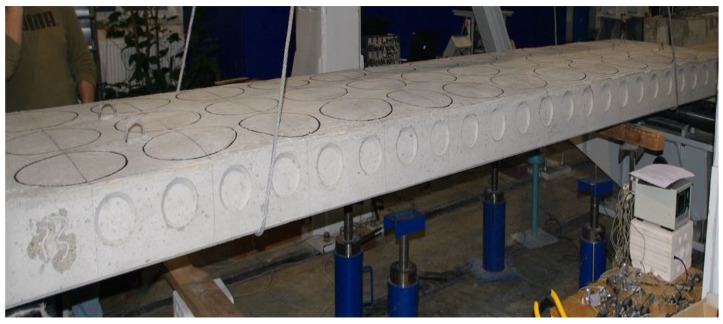
Overall view of the manufactured slab.

**Figure 4 materials-15-03013-f004:**
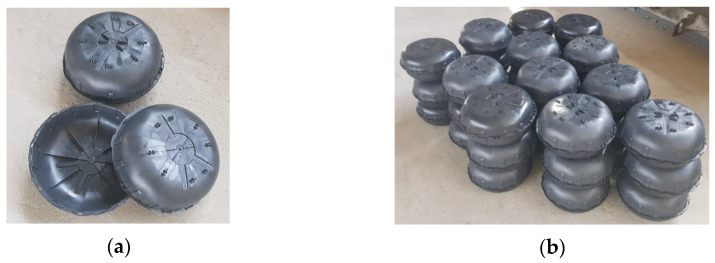
“Cobiax” plastic inserts. (**a**) Parts of unassembled inserts, (**b**) assembled inserts, prepared for concreting.

**Figure 5 materials-15-03013-f005:**
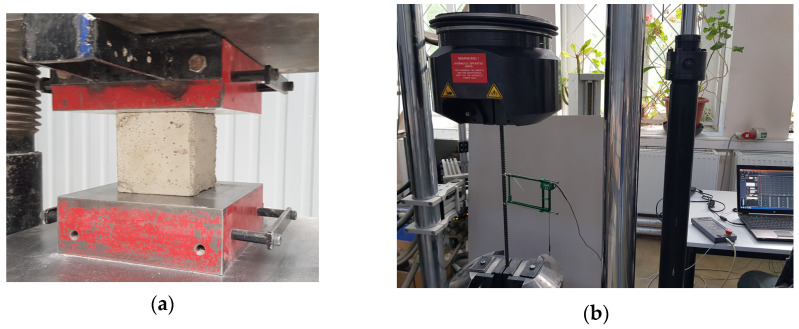
Determination of technical characteristics of the concrete and reinforcements. (**a**) Determination of the compressive strength of concrete cubes, (**b**) tensile strength of reinforcements.

**Figure 6 materials-15-03013-f006:**
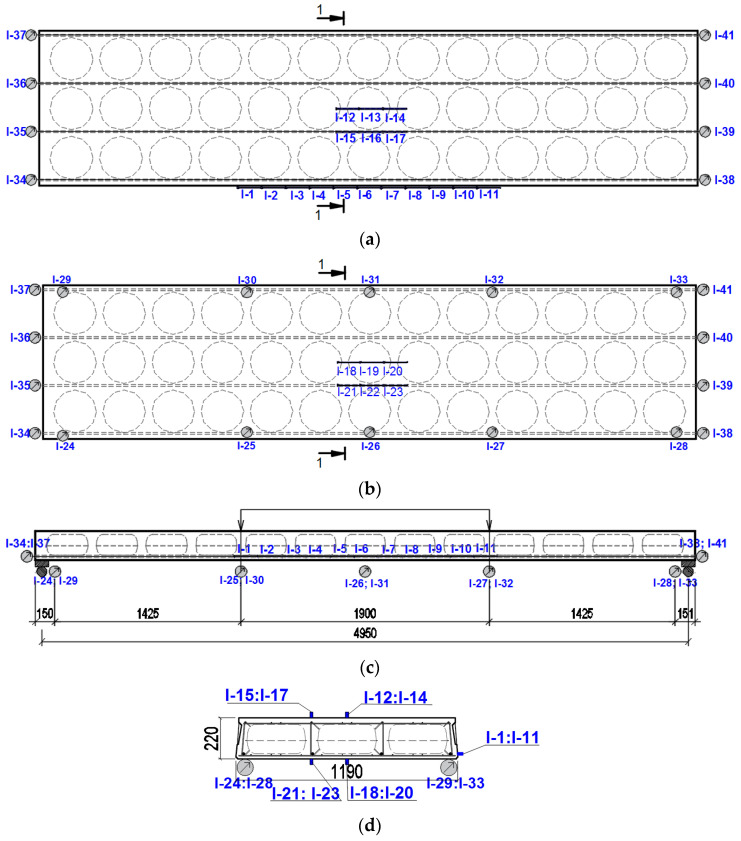
Slab test scheme with strain gauges. (**a**) The top of the slab; (**b**) the bottom of the slab; (**c**) side view of the slab; (**d**) slab section 1-1.

**Figure 7 materials-15-03013-f007:**
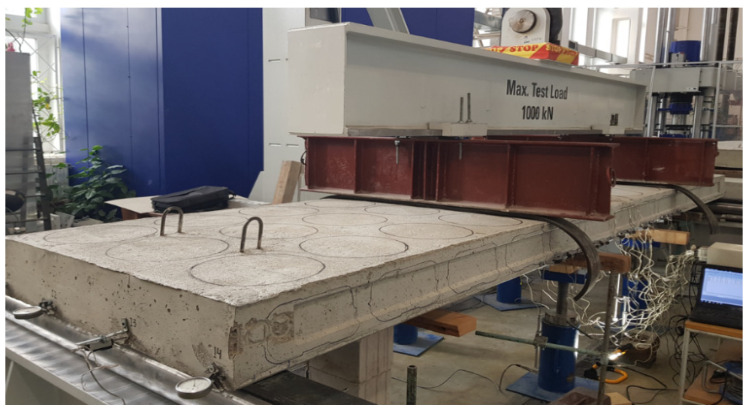
General view of the slab test.

**Figure 8 materials-15-03013-f008:**
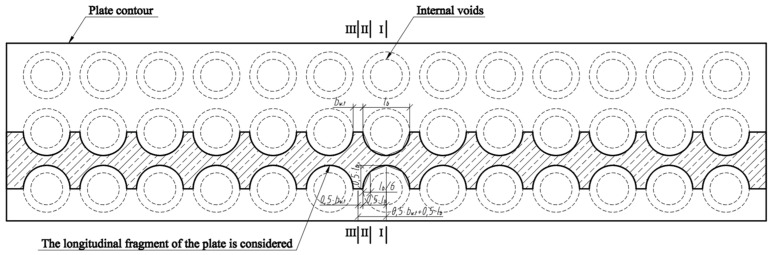
Slab scheme and sections.

**Figure 9 materials-15-03013-f009:**
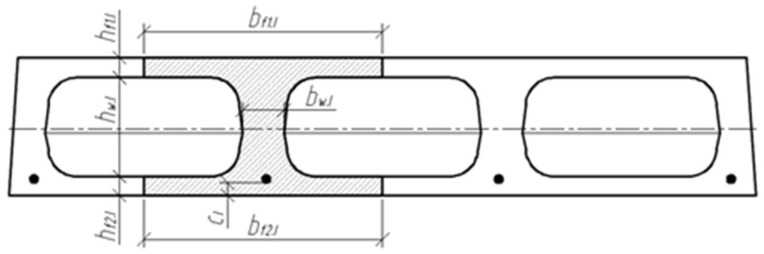
Slab cross-section I-I, where the perimeter of the void is greatest.

**Figure 10 materials-15-03013-f010:**
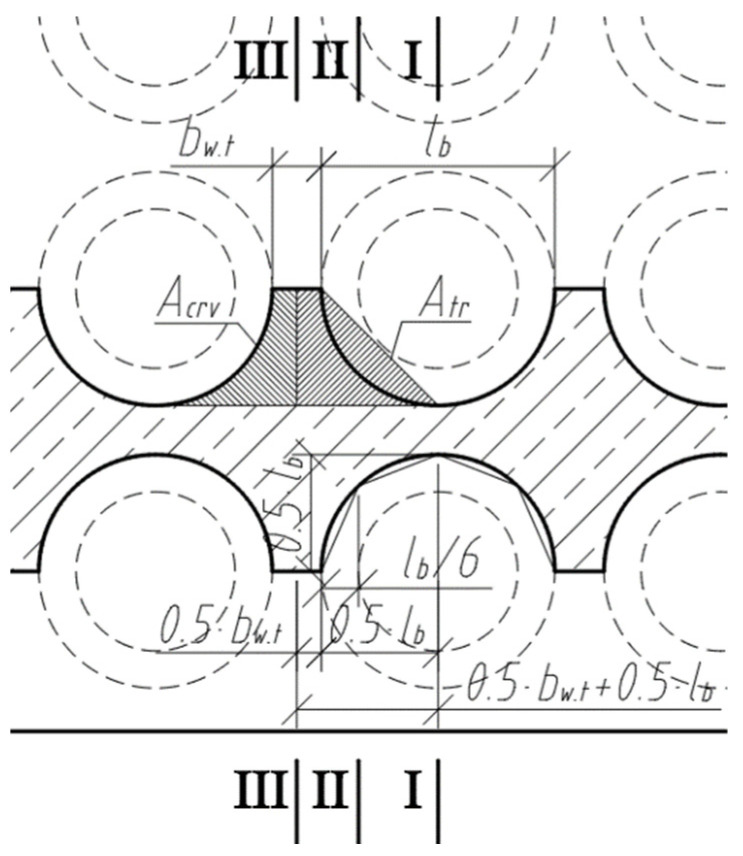
Scheme to determine the position of section II-II.

**Figure 11 materials-15-03013-f011:**
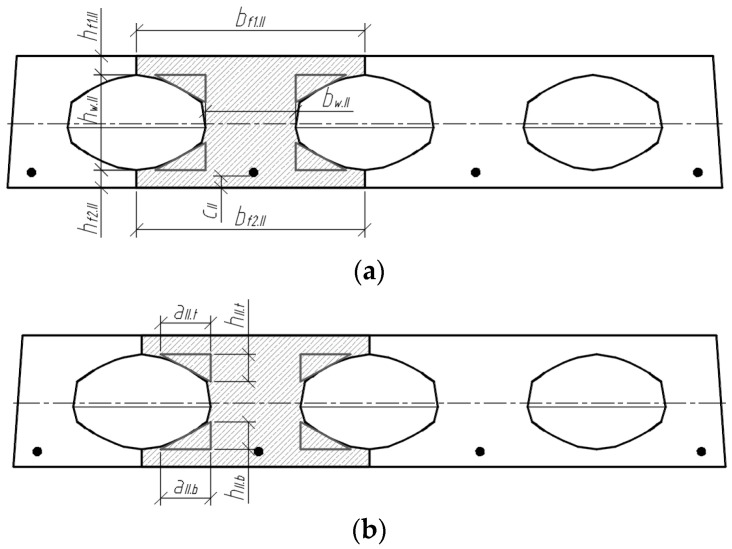
Analysed cross-section II-II; (**a**) edge parameters; (**b**) parameters of the triangular sections of the cross-section.

**Figure 12 materials-15-03013-f012:**
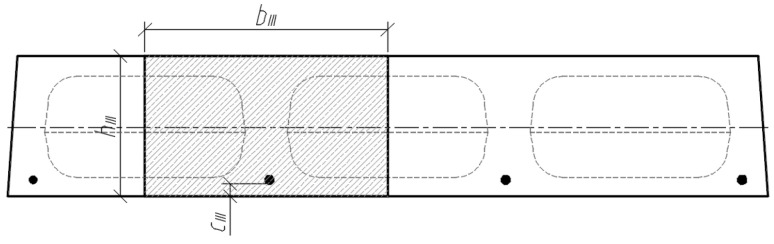
Analysed cross-section III-III.

**Figure 13 materials-15-03013-f013:**
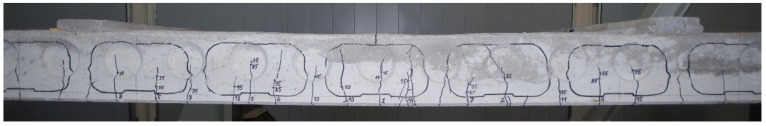
Distribution of cracks at voids and webs.

**Figure 14 materials-15-03013-f014:**
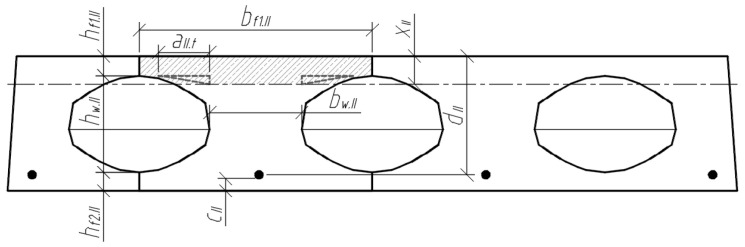
Cracked cross-section within section II-II, when $x_{II}>h_{f1.II}$.

**Figure 15 materials-15-03013-f015:**
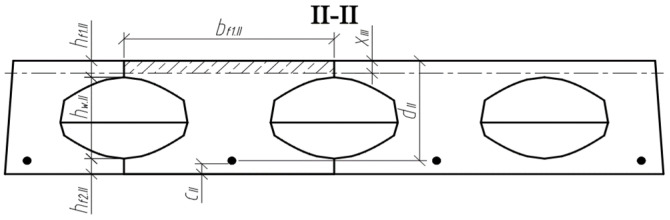
Cracked cross-section II-II, when $x_{II}\leq h_{f1.II}$.

**Figure 16 materials-15-03013-f016:**
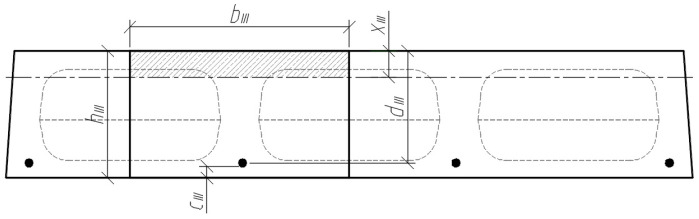
Cracked cross-section III-III.

**Figure 17 materials-15-03013-f017:**
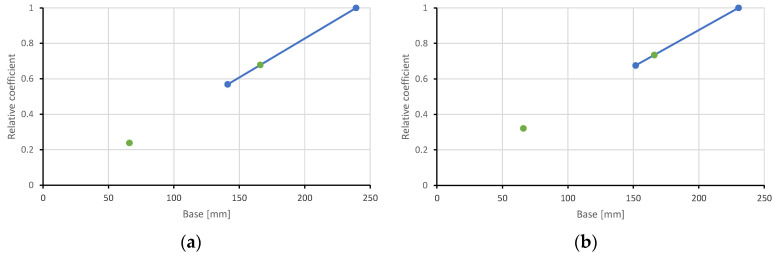
Relative coefficients for the prediction of the depth of the neutral axis in section II-II and III-III; (**a**) slab 1; (**b**) slab 2.

**Figure 18 materials-15-03013-f018:**
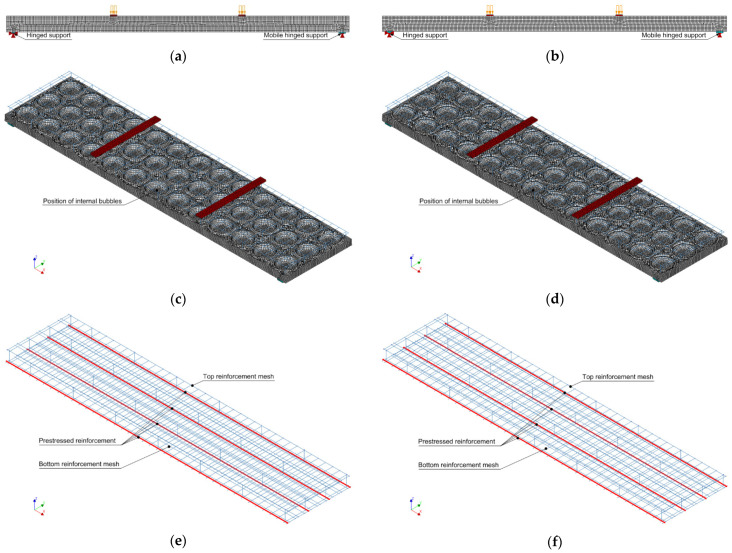
Views of modelled slabs: (**a**,**b**) supporting and loading of slabs I pl and II pl; (**c**,**d**) positions of voids in slabs 1 and 2; (**e**,**f**) reinforcement in the model of slab 1 and 2.

**Figure 19 materials-15-03013-f019:**
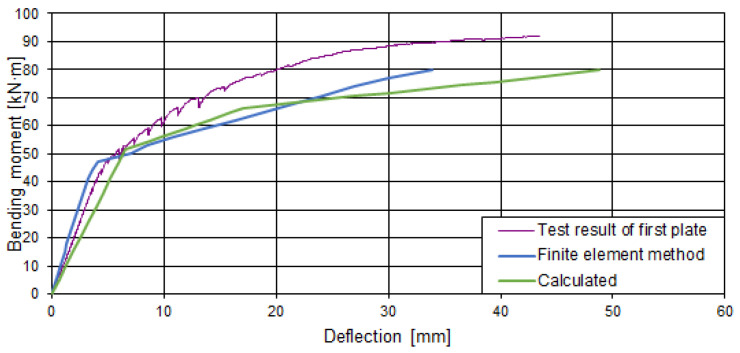
Comparison of the deflections of the first slab (I pl).

**Figure 20 materials-15-03013-f020:**
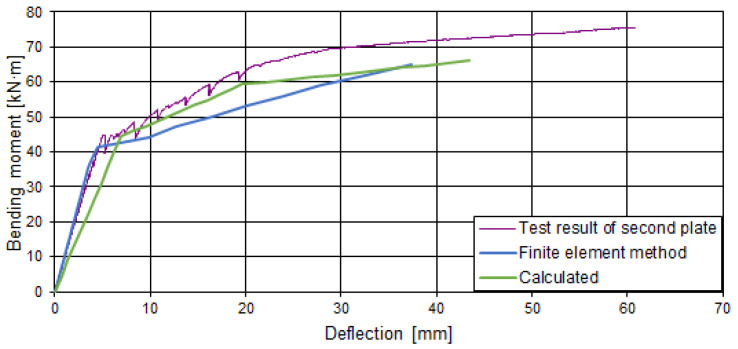
Comparison of the deflections of the second slab (II pl).

**Figure 21 materials-15-03013-f021:**
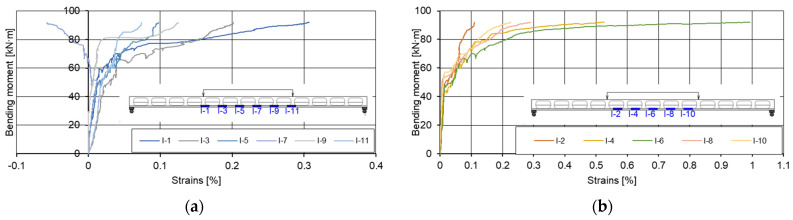
Graphs of bending moments: strains of the first slab (I pl) at the level of pretensioned reinforcement. (**a**) Strains between the plastic inserts and (**b**) strains at the place of the plastic inserts.

**Figure 22 materials-15-03013-f022:**
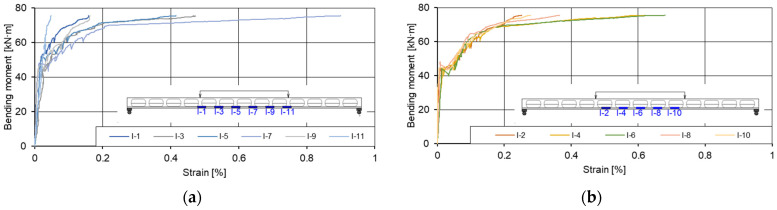
Graphs of bending moments: strains of the second slab (II pl) at the level of pretensioned reinforcement. (**a**) Strains between the plastic inserts and (**b**) strains at the place of the plastic inserts.

**Figure 23 materials-15-03013-f023:**
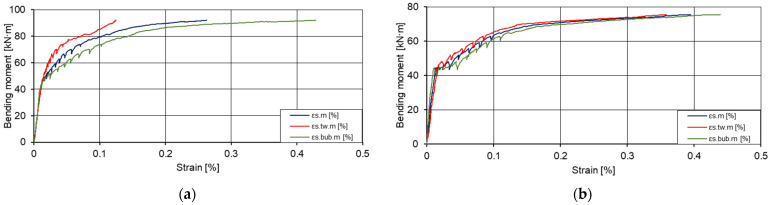
Graphs of bending moments: strains in the tensile zone of the slabs. (**a**) First slab (I pl); (**b**) second slab (II pl), where *ε*_*s*.*m*_ is the average strain [%]; *ε*_*s*.*tw*.*m*_ is the average strain values in the wall zone [%]; and *ε*_*s*.*bub*.*m*_ is the average strain values in the insert zone [%].

**Figure 24 materials-15-03013-f024:**
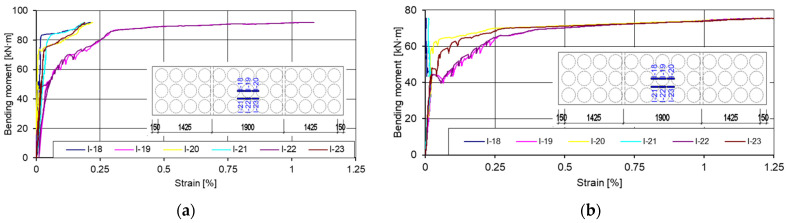
Graphs of bending moments: strains in the tensile zone of the slabs. (**a**) First slab (I pl); (**b**) second slab (II pl).

**Figure 25 materials-15-03013-f025:**
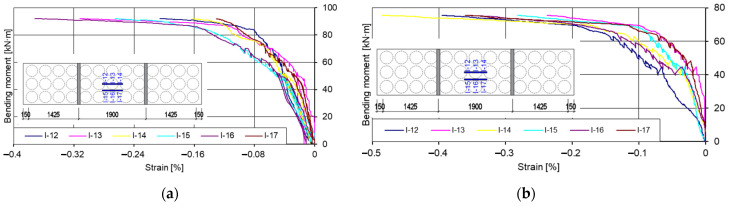
Graphs of bending moments: strains in the compressive zone of the slabs. (**a**) First slab (I pl); (**b**) second slab (II pl).

**Table 1 materials-15-03013-t001:** The technical data of reinforced concrete floor slabs.

Slab No.	Slab Length, mm	Average Slab Width, mm	Slab Height, mm	Number of Inserts, pcs.	Total Volume of Inserts in Slabs, m^3^	Concrete C30/37 Content, m^3^	Reinforcement Intensity % ^1^
I pl.	5100	1175	220	39	0.3682	0.95015	0.172
II pl	5100	1175	220	39	0.3682	0.95015	0.172

^1^ Reinforcement intensity was calculated for the slab’s total cross-sectional area without excluding the area of the voids.

**Table 2 materials-15-03013-t002:** Technical characteristics of inserts [21].

Line No.	The Name of the Insert Parameter	Unit	Size Value
1	Diameter of insert	mm	315
2	Height of insert	mm	160
3	Weight of insert	g	379.2
4	Internal volume of insert	m^3^	0.00944

**Table 3 materials-15-03013-t003:** Concrete technical characteristics.

Slab No	*f*_*c*,*cub*,100_ ^1^ [MPa]	*f*_*c*,*cub*,150_ ^2^ [MPa]	*f_cm_* ^3^ [MPa]	*f_ct_* ^4^ [MPa]	*E_c_* ^5^ [GPa]
I pl	44.00	41.80	31.86	2.49	30.44
II pl	44.59	42.36	32.24	2.51	30.54

^1^ Average concrete cube compressive strength, when the cube’s dimensions were 100 × 100 × 100 mm; ^2^ Average concrete cube compressive strength, when the cube’s dimensions were 150 × 150 × 150 mm; ^3^ Average concrete cylindrical compressive strength; ^4^ Concrete tensile strength; ^5^ Concrete elastic modulus.

**Table 4 materials-15-03013-t004:** Characteristics of pretensioned reinforcements.

ø [mm]	*A_s_* [m^2^] ^1^	*f*_*y*,02_ [MPa] ^2^	*E_s_* [GPa] ^3^	*σ_p*,*I_* [MPa] ^4^	*σ_p*,*II_* [MPa] ^5^
12	0.000111436	869.72	182.83	372.3	373.7

^1^ Bar cross-sectional area; ^2^ Reinforcement strength according to yield point; ^3^ Reinforcement elastic modulus; ^4^ Stresses in pretensioned reinforcement bars of the first slab; ^5^ Stresses in pretensioned reinforcement bars of the second slab.

## Data Availability

The data presented in this study are available on request from the corresponding author.

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
