# Peer review of "Deflection Estimation Model for Prestressed Concrete Slabs with Plastic Inserts Forming Voids"

_materials, 2022, doi:10.3390/ma15093013_

Round 1

Reviewer 1 Report

The reviewer thanks the authors for the work done. The technical contents of the paper are interesting and the findings are useful information for field applications in the future. Nevertheless, the publication in the “Materials, MDPI” is not recommended unless the following suggestions are taken into account within the article:

1) The goal of the work is to numerically calibrate measurements for providing information in terms of vertical displacements and load-carrying capacity of Prestressed Concrete (PC) plates with plastic forming voids. Similar calibrations were performed in the literature regarding PC beam members. Please, refer to this issue, and cite the following references in the Introduction:
-  Damage detection in a post tensioned concrete beam – Experimental investigation, Eng. Struct. 128 (2016) 15–25.
-  Influence of prestressing on the behavior of uncracked concrete beams with a parabolic bonded tendon, Struct. Eng. Mech. 77 (1) (2021) 1–17.

2) The authors did not mention about the prestressing state of the PC plates tested. How much prestressing loading has been applied to the PC plates before testing ? How much prestressing loading has been measured in the PC plates during and after testing ? Please, specify.

3) Please, provide the frequency (or the period) of the recording data by all the equipment and devices used.

4) Please, provide the technical characteristics of all the equipment and devices used.

5) Please, provide the standards of compressive and tensile tests performed on concrete and prestressing strands (Figure 5).

6) Section 4. The section should furnish more information regarding the types of finite elements (FE) adopted with corresponding amount and mesh sizes.

7) Section 5. The calibrations between experimental and FE measurements can be improved by the following indications:
a) The nonlinear FE static analyses should consider the prestressing loads applied in the form of tensile forces in the strands with the aim to take the second-order effects into account.
b) The nonlinear FE static analyses should consider a reduced value of the concrete elastic modulus for taking into account the early-age creep effects. Particularly, the reviewer suggests the following reference for this particular aspect: “RILEM draft recommendation: TC-242-MDC multi-decade creep and shrinkage of concrete: material model and structural analysis. Model B4 for creep, drying shrinkage and autogenous shrinkage of normal and high-strength concretes with multi-decade applicability. Mater. Struct. 2015 48 (4) 753–70.”
c) The experimental vertical displacements at the supports of the PC plates should proportionally be subtract from those measured along the spans.

8) With the aim to define a numerical formula for calculating vertical displacements  of PC plates with plastic forming voids, the reviewer suggests the following reference for this development which, in turn, was implemented for PC beam members: "Feasibility study of prestress force prediction for concrete beams using second–order deflections, Int. J. Struct. Stab. Dy. 18 (10) (2018) 1–19."

9) Based on the aforementioned comments, the current state of knowledge relating to the manuscript topic should be covered and clearly presented, with the authors’ contributions too. In this regard, the authors should make their effort to address these issues adding additional comments on the state of the art and the proposed aspects.

10) I suggest to the authors to edit all the text of the article with the help of a native English speaker. Grammar, punctuation, spelling, verb usage, sentence structure, conciseness, readability and writing style can also be improved.

Author Response

Please find answers attached.

Reviewer 2 Report

Dear authors. Congratulations on your interesting research. I have some comments that will improve the quality of the manuscript.

(1) English is not my first (native) language, but I believe that the manuscript needs professional linguistic proofreading. Examples:

  • you commonly use the term "plate", it should rather be "slab", there is also the term "boards" (line 198)
  • what does the term "continuous plate" mean? is it "solid slab"?
  • I have always been taught that the term "prestressed" refers rather to concrete (e.g. prestressed concrete), we use prestressing reinforcement,
  • what does "more reduced mix" (line 45?), previous cracking (line 50) mean, - was the bending load transmitted rather than the "compresive load" (line 215)?

(2) information about the reinforcement is insufficient (rebar spacing is missing) - Figure 1,

(3) How was the "Reinforcement intensity" calculated (specified in Table 1)?

(4) The material characteristics shown in Tables 3 and 4 are insufficient (strains are missing), their relationship with the material models of the numerical model is not shown (concrete model, elastic-plastic steel model)

(5) the properties of mild steel are not shown,

(6) the numbering of the formulas is missing

(7) the calculation model does not contain information on how to calculate the cracking moment

(8) as in the formula for curvature (line 308), take into account the prestressing effect - it is a long-term load

(9) What model of concrete was used in the numerical analysis?

(10) the FEM model ignores the creep, were the losses of the prestressing force, e.g. resulting from partial relaxation, taken into account?

(11) Authors wrote that "the proposed method ... showed a sufficiently good agreement with the numerical analysis", but the differences for the laboratory model are much greater, there is no discussion of the reasons.

Author Response

Please find answers attached.

Round 2

Reviewer 1 Report

The authors carried out the required revisions, and the contribution of their work deserves to publish in the "Materials, MDPI".

Reviewer 2 Report

The authors incorporated all my comments and answered all my questions. I have no further comments. The manuscript may be published as is.
I do not like the large discrepancy in the results of the deflection calculations (Figure 20). In future studies, the authors should elaborate more on this issue. Perhaps in formula (22) coefficients similar to the distribution factor in Eurocode 2 can be introduced (allowing to take into account tension stiffening effect etc)